# Suspected Human-to-Cat Spillover of SARS-CoV-2 Omicron Variant in South Korea

**DOI:** 10.3390/v16071113

**Published:** 2024-07-11

**Authors:** Ju-Hee Yang, Yeonsu Oh, Sung-Hyun Moon, Gun-Hee Lee, Jae-Young Kim, Yeon-Kyung Shin, Dongseob Tark, Ho-Seong Cho

**Affiliations:** 1Korea Zoonosis Research Institute, Jeonbuk National University, Iksan 545431, Republic of Korea; juheeppuing@jbnu.ac.kr (J.-H.Y.); ghlee80@jbnu.ac.kr (G.-H.L.); 2College of Veterinary Medicine and Institute of Veterinary Science, Kangwon National University, Chuncheon 24341, Republic of Korea; yeonoh@kangwon.ac.kr; 3College of Veterinary Medicine and Bio-Safety Research Institute, Jeonbuk National University, Iksan 54596, Republic of Korea; chunsu17@naver.com; 4Tae Neung Animal Hospital, Seoul 02033, Republic of Korea; idkjyah@hanmail.net; 5Foreign Animal Disease Division, Animal and Plant Quarantine Agency, Gimcheon 39660, Republic of Korea; shinyk2009@korea.kr

**Keywords:** SARS-CoV-2, Omicron variant, spillover, cat

## Abstract

This retrospective study reports the isolation and characterization of Severe Acute Respiratory Syndrome coronavirus 2 (SARS-CoV-2) from a household cat in South Korea. The cat, which was presented with respiratory symptoms, was identified during a retrospective analysis of samples collected between April 2021 and March 2022. Genomic sequencing revealed that the isolated virus belonged to the Omicron variant (BA.1), coinciding with its global emergence in early 2022. This case study provides evidence for the potential of direct human-to-cat transmission of the Omicron variant in South Korea during its period of widespread circulation. Our findings underscore the importance of continuous monitoring of SARS-CoV-2 in both human and animal populations to track viral evolution and potential spillover events.

## 1. Introduction

The COVID-19 pandemic, caused by the novel coronavirus, Severe Acute Respiratory Syndrome Coronavirus 2 (SARS-CoV-2), has had a profound impact on global health [1]. While initially considered a zoonotic disease [2], SARS-CoV-2 has demonstrated a capacity for reverse zoonosis, with documented transmission from humans to various animal species, including non-human primates, wild animals, and domestic pets [3]. The outbreaks of the host jump transmission (i.e., human-to-animal spillover) of SARS-CoV-2 have been reported in various animals and countries [4,5,6]. Studies have recently revealed that domestic animals living in households with COVID-19-positive individuals can also test positive for SARS-CoV-2. The majority of household animals, primarily cats and dogs, exhibited mild respiratory symptoms like coughing and sneezing [7]. Cats, in particular, have shown susceptibility to SARS-CoV-2 infection, often exhibiting mild respiratory symptoms [8].

By 25 January 2022, more than half of the COVID-19 patient samples in South Korea were attributed to the SARS-CoV-2 lineage B.1.1.529, commonly known as the Omicron variant [9]. The emergence of novel SARS-CoV-2 variants, such as the Omicron variant (lineage B.1.1.529), raised concerns about their potential to infect animal populations. However, as of early 2022, there was no documented transmission of SARS-CoV-2 Omicron variants from humans to animals in South Korea.

To address this knowledge gap, we conducted a retrospective epidemiological study examining cases of SARS-CoV-2 in cats from April 2021 to March 2022. Cats visiting local animal hospitals and shelters during this period were randomly selected for inclusion in the study.

This study reports a confirmed case of SARS-CoV-2 Omicron variant infection in a domestic cat in South Korea, representing the first documented case during this period. We discuss the implications of this finding for ongoing public monitoring and surveillance efforts.

## 2. Materials and Methods

### 2.1. Study Design and Sample Collection

This study was conducted between April 2021 and March 2022. A total of 546 cats were randomly selected from local animal hospitals and animal shelters across South Korea. The demographic data of the cats were recorded, including breed, sex, age, recent medical history, and the presence of respiratory signs (cough, sneezing, nasal discharge, and conjunctivitis). Nasopharyngeal/oropharyngeal swabs and blood samples were collected from each cat. Swabs were placed in a universal transport medium (UTM) (GDL Korea, Seoul, Republic of Korea) and transported on ice to the laboratory, where they were stored at −20 °C until analysis. Blood samples were clotted, prepared as serum aliquots, and stored at −20 °C until analysis.

### 2.2. RNA Extraction and Reverse Transcription Real-Time qPCR

All swab samples were processed to detect the presence of SARS-CoV-2 RNA, following the outlined protocol. The UTMs were thoroughly mixed to produce 200 μL of aliquots necessary for RNA extraction. This RNA was isolated utilizing a 16TU-CV19 Viral DNA/RNA Preparation Kit (MiCo BioMed, Seoul, Republic of Korea) and a Veri-Q PREP M16 device (MiCo BioMed, Seoul, Republic of Korea), adhering strictly to the guidelines provided by the manufacturer. A reverse-transcription real-time qPCR assay was then performed to identify the ORF3a and nucleocapsid (N) genes of SARS-CoV-2, employing a proprietary nCoV-QM PCR kit (MiCo BioMed) with a specifically designed instrument, Veri-Q PCR 316 QD-P100 (MiCo BioMed). The PCR setup involved a 10 μL reaction volume comprising 3 μL of master mix (including polymerase, reverse transcriptase, buffer, and stabilizer), 1 μL of the primer/probe mix, 1 μL of internal positive control, and 5 μL of the extracted RNA. The PCR cycling parameters included a reverse transcription step at 50 °C for 10 min, an initial denaturation at 95 °C for 3 min, followed by 45 cycles of denaturation at 95 °C for 9 s, and annealing and extension at 58 °C for 30 s each.

### 2.3. Serological Analysis

Serological assays targeting the nucleocapsid protein (N protein) of SARS-CoV-2 were conducted using a commercially available indirect ELISA (ID Screen^®^ SARS-CoV-2 Double Antigen Multi-species, IDvet, Grabels, France). This procedure involved the addition of serum samples and horseradish peroxidase (HRP) conjugated N protein-recombinant antigen to microwell plates that had been pre-coated with a purified N protein-recombinant antigen. The detection of SARS-CoV-2 antibodies in the sera was confirmed by measuring the optical density (OD) at 450 nm. The assay was considered valid if the optical density of the positive control (ODPC) was ≥0.35 and the ratio of the optical density between the positive and negative controls (ODPC/ODNC) exceeded three. The optical density for each test sample (ODN) was utilized to compute the S/P ratio (expressed as percentage) where S/P = 100 × (ODN − ODNC)/(ODPC − ODNC). An ELISA result was deemed positive if the S/P ratio exceeded 60%, ambiguous if it fell between 50% and 60%, and negative if it was below 50% [10].

### 2.4. Virus Isolation and Propagation

To isolate the infectious SARS-CoV-2 virus from the JBNU-cat sample, a nasopharyngeal swab sample was inoculated onto Vero E6 cells (African green monkey kidney epithelial cells; ATCC^®^ CRL-1586^TM^), known to be permissive to SARS-CoV-2 infection [11]. Cells were maintained in Dulbecco’s Modified Eagle Medium supplemented with 10% fetal bovine serum and 1% penicillin-streptomycin at 37 °C in a humidified atmosphere containing 5% CO_2_. Following inoculation, the cells were monitored daily for cytopathic effects (CPE) indicative of viral replication. The appearance of CPE, characterized by cell rounding, detachment, and syncytia formation (Figure 1B), was considered preliminary evidence of successful virus isolation. To confirm the presence of infectious virus, supernatant from the inoculated Vero E6 cells exhibiting CPE was collected and passaged onto fresh Vero E6 cell monolayers. The development of CPE in subsequent passages confirmed the presence of a replicating virus. Viral titers were determined using 50% tissue culture infective dose (TCID_50_/mL).

### 2.5. Whole-Genome Sequencing and Phylogenetic Analysis

Viral RNA was extracted from the JBNU-cat SARS-CoV-2 isolate using Trizol LS (Invitrogen, Carlsbad, CA, USA). cDNA was synthesized from the extracted RNA using the Sequence-Independent, Single-Primer-Amplification (SISPA) method for whole genome sequencing [12]. A sequencing library was prepared using custom-designed primer sets and the Illumina platform-based BTSeq SARS-CoV-2 whole-genome sequencing (WGS) kit (Celemics, Inc., Seoul, Republic of Korea) according to the manufacturer’s instructions. The prepared library was sequenced on the Illumina iSeq 100 platform, generating paired-end reads with a length of 2 × 150 bp.

Raw sequencing reads were quality-controlled by trimming adapter sequences, removing dual indices, and filtering out low-quality reads. The remaining high-quality reads were aligned to the reference genome of hCoV-19/Wuhan/WIV04/2019 (EPI_ISL_402124) using CLC Genomics Workbench 8.0. A consensus genome sequence for the JBNU-cat/2022 strain was generated, which spanned 29,841 nucleotides and had an average coverage depth of 14,783×. The sequences of hCoV-19/South Korea/JBNU-cat/2022 were deposited in the GISAID database (https://www.gisaid.org) (accession numbers EPI_ISL_16577826). To analyze the phylogenetic relationship, a total of 121 SARS-CoV-2 patient samples collected from December 2021 to November 2022, along with the SARS-CoV-2 isolate from JBNU-cat, were included.

## 3. Results

### 3.1. Regional Screening and Isolation of SARS-CoV-2 from a Domestic Cat

This study investigated the potential for SARS-CoV-2 transmission to domestic cats in South Korea. Regional screening efforts, encompassing both antigen and antibody testing, revealed the presence of SARS-CoV-2 in one feline sample, designated as JBNU-cat (Table 1 and Table 2, and Figure 1A).

Following the initial identification, the JBNU-cat SARS-CoV-2 virus was successfully isolated and propagated in Vero E6 cells. This isolation was confirmed through the visualization of viral antigens within infected cells using fluorescence microscopy (Figure 1C,D). The isolated virus exhibited robust replication competence, achieving high viral titers and demonstrating the capacity for continuous passage in Vero E6 Cells, as evidenced by immunofluorescence assay data.

To further confirm the virus’s stability and replication competence during passage, we assessed that the viral copy number of the JBNU-cat isolate was comparable to the positive control of the second passaged Omicron BA.1 (NCCP43408) in the first passage, indicating robust replication competence. Subsequent passages showed an increase in the viral copy number up to the third generation, further confirming efficient viral replication. CPE, characterized by cells, became rounded or irregular in shape and detached, and it was observed in all passages, confirming viral proliferation. While we focused on the first few passages to confirm consistent replication, continuous CPE and high viral titers were observed in later passages, supporting the claim of continuous propagation without loss of viral characteristics (Table 3).

### 3.2. Phylogenetic Analysis and Spike Protein Characterization of JBNU-Cat SARS-CoV-2

To determine the evolutionary relationship of the isolated virus, phylogenetic analysis was conducted. Results revealed that the JBNU-cat SARS-CoV-2 isolate clusters within lineage BA.1.1, according to the GISAID classification system. This lineage falls within the GRA clade, a grouping within GISAID based on shared ancestry that corresponds to the B.1.1.529.2 lineage within the Pangolin classification system. This classification firmly places the JBNU-cat isolate within the Omicron variant lineage [13] (Figure 2).

Further characterization focused on the viral spike protein, a critical determinant of viral infectivity. Comparative analysis of the JBNU-cat spike protein sequence with that of the original SARS-CoV-2 isolate and other human-derived variants of concern, including Omicron and Delta, revealed multiple mutation sites (Figure 3). Despite these mutations relative to Hu-1, Delta, and Omicron variants, the overall structural conformation of the JBNU-cat spike protein remains largely consistent with that of the human Omicron variant. This similarity likely stems from the conservation of key structural elements within the spike protein, such as those involved in binding to the ACE2 receptor and mediating membrane fusion. This structural conservation suggests the potential for retained infectivity and underscores the need for further investigation.

## 4. Discussion

This research, documenting the isolation and characterization of the SARS-CoV-2 Omicron variant (BA.1.1 lineage) from a domestic cat in South Korea, contributes significantly to the growing body of evidence regarding the zoonotic potential of SARS-CoV-2 [14]. While our findings are based on data collected between April 2021 and March 2022, they highlight the importance of continuous monitoring for SARS-CoV-2 in animal populations, particularly following a period of significant human transmission.

Beyond confirming feline susceptibility, this retrospective case study prompts a deeper examination of the complex interplay between viral evolution, cross-species transmission, and public health in the context of a rapidly evolving pandemic.

The high degree of homology between the human and feline ACE2 (fACE2) receptors, a critical determinant of viral entry, has been well-established. Feline ACE2 shares 85.2% homology with its human counterpart (hACE2) [15]. This case study reinforces this understanding, demonstrating the successful isolation and propagation of the Omicron variant in Vero E6 cells, a cell line known to express the human ACE2 receptor. The observed robust replication competence, evidenced by high viral titers and continuous passaging ability, further underscores the permissive nature of feline cells for SARS-CoV-2 infection and potential for sustained viral replication within feline hosts [16].

The phylogenetic analysis, placing the JBNU-cat isolate within the human Omicron BA.1.1 lineage, provides compelling evidence for human-to-cat transmission. The JBNU-cat sample was collected in February 2022 from a cat residing in Jeonju, Korea. During this period, Korea experienced a significant surge in human cases of the SARS-CoV-2 Omicron variant, with 36.7% of sequenced cases attributed to this variant [17]. The close temporal relationship between the peak in human Omicron cases and the detection of the Omicron variant in JBNU-cat supports the likelihood of human-to-cat spillover. While contact with other infected animals cannot be definitively ruled out, the cat’s owner in the case tested positive for the Omicron variant. While the precise directionality of transmission events is often challenging to ascertain, the predominance of the Omicron variant in the human population at the time of this study suggests spillover from humans to cats. This finding aligns with previous reports of SARS-CoV-2 transmission from humans to various animal species, including minks, white-tailed deer, and hamsters, highlighting the broad host range of this virus and the potential for the establishment of cryptic transmission cycles in animal populations [8].

The identification of multiple mutations in the spike protein of the JBNU-cat isolate, despite maintaining a similar overall structure to the human Omicron variant, raises intriguing questions about viral adaptation. While the functional consequences of these mutations require further investigation, their presence suggests potential selective pressures exerted by the feline host environment. These adaptations could potentially influence viral infectivity, transmissibility, and even pathogenicity in cats, underscoring the need for continuous genomic surveillance of SARS-CoV-2 in animal populations to monitor for the emergence of novel variants with altered phenotypic characteristics [18].

The detection of this Omicron variant in a domestic cat in South Korea, a country that has experienced a significant Omicron wave in the human population, highlights the importance of strengthening One Health-based surveillance efforts to detect and monitor the potential emergence of SARS-CoV-2 reservoirs in animal populations [17,19].

It is noteworthy that only one out of 546 cats sampled tested positive for the novel coronavirus. While this finding confirms the susceptibility of cats to this virus, the low detection rate limits our ability to draw definitive conclusions about its prevalence and transmission dynamics within feline populations.

Of the 546 cats included in the study, all exhibited at least one COVID-19-related clinical sign, such as fever (body temperature above 39.5 °C) and cough. However, only 19 cats (3.48%) tested positive for SARS-CoV-2 antibodies, indicating no direct correlation between serological status and clinical presentation. This discrepancy suggests that the observed clinical signs were likely attributable to other factors, such as infection with other respiratory pathogens or allergies, which are common in cats. Furthermore, the majority of seropositive cats (18 out of 19) were negative for viral RNA by qPCR, indicating prior exposure and viral clearance rather than active infection. Notably, the single cat positive for viral RNA (case no. 17) also exhibited fever and cough, suggesting active infection in this case.

This study’s findings have profound implications for public health, particularly in the context of a One Health approach to pandemic preparedness and response. The documented case of cat-to-human transmission in Thailand serves as a stark reminder of the bidirectional nature of zoonotic transmission and the potential for spillback from animal reservoirs to humans [20]. This emphasizes the need for integrated surveillance systems that encompass both human and animal populations to effectively monitor viral evolution and transmission dynamics.

Furthermore, the potential for SARS-CoV-2 to establish enzootic or epizootic cycles in animal populations poses a significant challenge for long-term pandemic control. Persistent viral circulation in animal reservoirs could lead to recurrent spillover events, potentially introducing novel variants into the human population and complicating ongoing vaccination efforts [21]. Therefore, developing and implementing effective mitigation strategies, such as the vaccination of susceptible animal populations, will be crucial for interrupting transmission chains and reducing the risk of future zoonotic outbreaks.

In conclusion, this study provides compelling evidence for the zoonotic potential of the SARS-CoV-2 Omicron variant and highlights the complex interplay between viral evolution, cross-species transmission, and public health. The findings underscore the need for a One Health approach that integrates surveillance, research, and intervention strategies across human and animal health sectors to effectively combat the pandemic and mitigate future zoonotic threats.

## Figures and Tables

**Figure 1 viruses-16-01113-f001:**
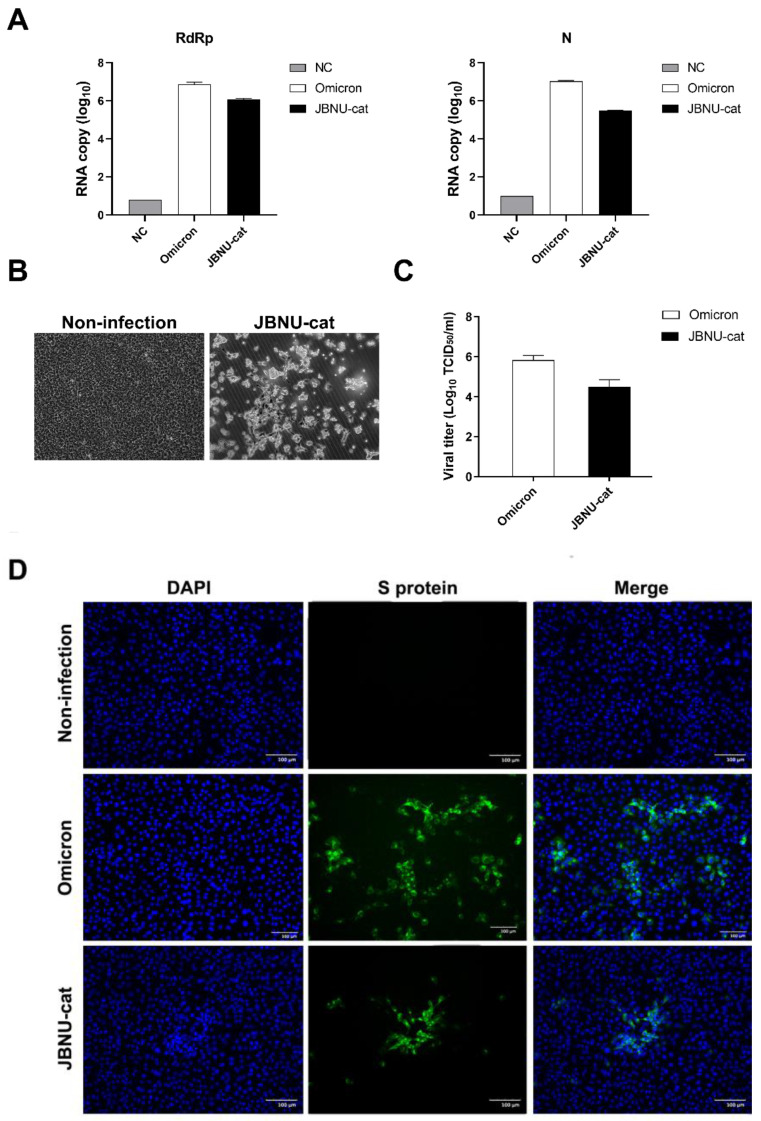
Determine the viral burden and infectious virus titer of SARS-CoV-2 isolated from cat (JBNU-cat). The copy number of N and RdRp gene of JBNU-cat measured by qRT-PCR. NC: Non-infected Vero E6, Omicron: SARS-CoV-2 Omicron BA.1 (NCCP43408, Korea National Culture Collection for Pathogens), JBNU-cat: SARS-CoV-2 virus isolated from JBNU-cat (**A**). Non-infected Vero E6 cells and a Cytopathic effect (CPE) in Vero E6 infected with JBNU-cat after second passages (**B**). The titration of infectious virus in Vero E6 (**C**). Vero E6 cells were infected with 0.1 MOI of Omicron variant or JBNU-cat for 1 h and detected spike (S) protein of SARS-CoV-2 (green signal) and nuclear staining with DAPI after 48 h (**D**) bar, 100 µm.

**Figure 2 viruses-16-01113-f002:**
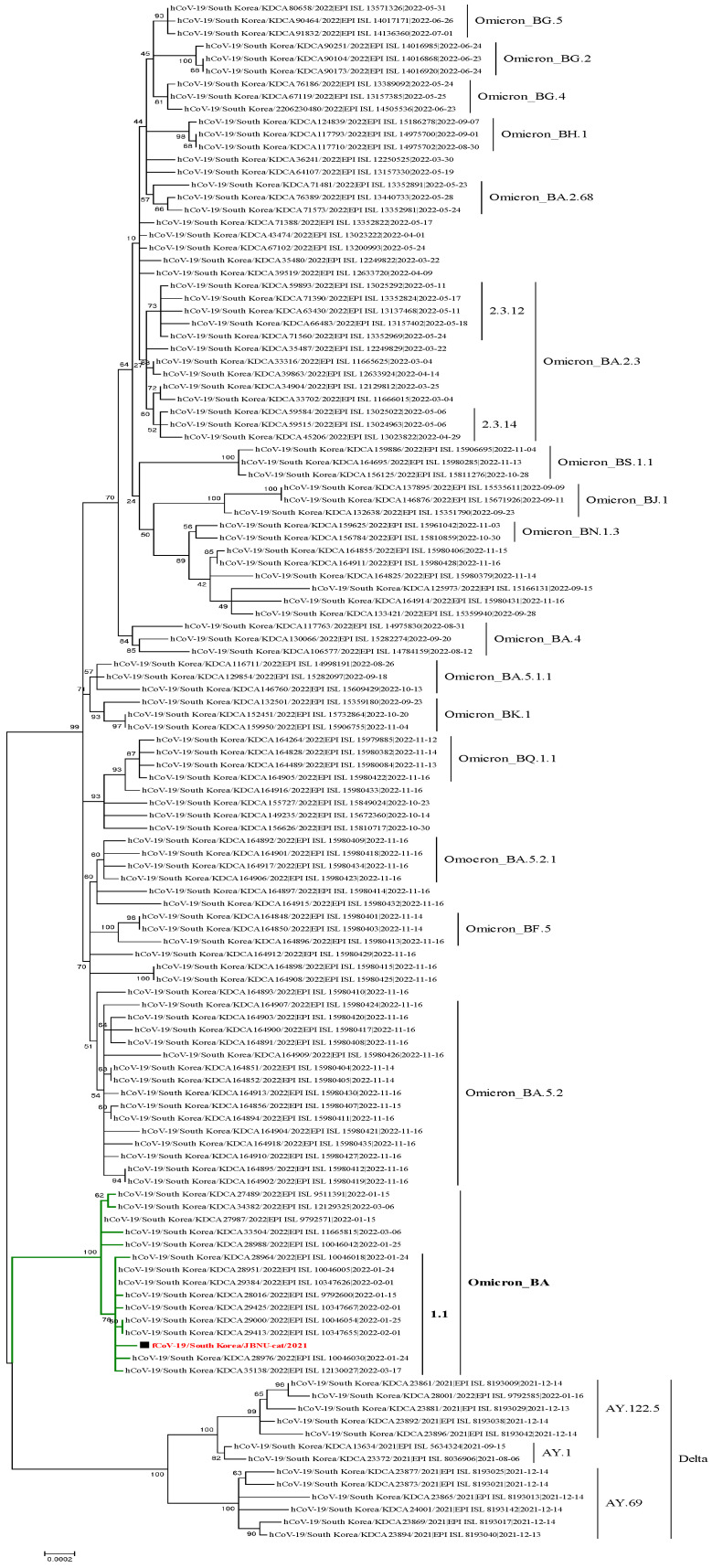
Phylogenetic tree of the JBNU-cat virus isolate and human SARS-CoV-2 sequences. The JBNU-cat viral genome is highlighted in red, while human SARS-CoV-2 genomes are shown in black. The tree was constructed using the Maximum likelihood method with 1000 bootstrap replicates in the Molecular Evolutionary Genetics Analysis (MEGA) v11.0 software, based on multiple sequence alignments generated using MAFFT v7.

**Figure 3 viruses-16-01113-f003:**
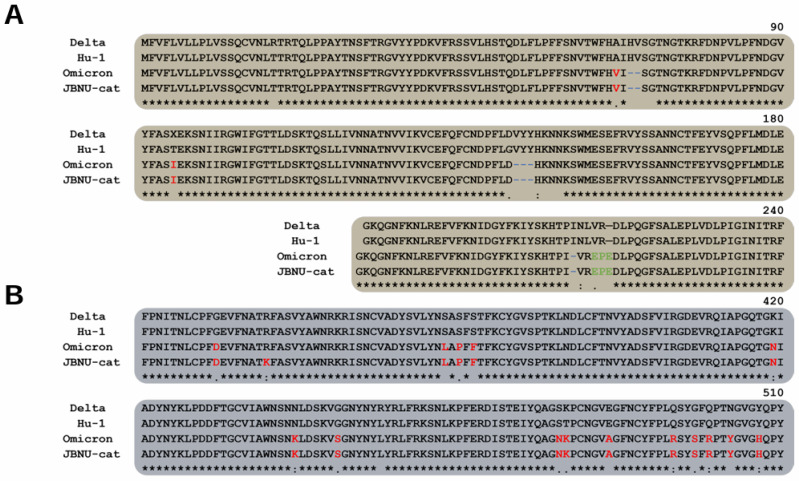
Spike protein alignment of the SARS-CoV-2 variants of concern (VOC). The protein sequence alignment with structural elements of the SARS-CoV-2 VOC N-terminal domain (NTD, (**A**)) and receptor-binding domain (RBD, (**B**)). The residues indicated, red; mutation, blue; deletion, green; insertion of residues.

**Table 1 viruses-16-01113-t001:** Prevalence of SARS-CoV-2 in cats by regions in Korea.

Province	Type	qPCR (%)	ELISA (%)
Seoul	Households	0/74 (0.00)	4/74 (5.41)
	Animal Shelters	0	0
Gyeonggi	Households	0/43 (0.00)	2/43 (4.65)
	Animal Shelters	0/38 (0.00)	1/38 (2.63)
Gangwon	Households	0/27 (0.00)	1/27 (3.70)
	Animal Shelters	0	0
Chungbuk	Households	0/35 (0.00)	2/35 (5.71)
	Animal Shelters	0	0
Chungnam	Households	0/26 (0.00)	1/26 (3.85)
	Animal Shelters	0/21 (0.00)	0/21 (0.00)
Gyeongbuk	Households	0/37 (0.00)	1/37 (2.70)
	Animal Shelters	0/33 (0.00)	2/33 (6.06)
Gyeongnam	Households	0/41 (0.00)	1/41 (2.44)
	Animal Shelters	0	0
Jeonbuk	Households	1/35 (2.86)	2/35 (5.71)
	Animal Shelters	0/35 (0.00)	0/35 (0.00)
Jeonnam	Households	0/37 (0.00)	2/37 (5.41)
	Animal Shelters	0/43 (0.00)	0/43 (0.00)
Jeju	Households	0/21 (0.00)	0/21 (0.00)
	Animal Shelters	0	0
Subtotal	Households	1/376 (0.27)	16/376 (4.26)
	Animal Shelters	0/170 (0.00)	3/170 (1.76)
Total		1/546 (0.18)	19/546 (3.48)

**Table 2 viruses-16-01113-t002:** Detection of antigens and antibodies against SARS-CoV-2 in cats by qPCR, ELISA, and PRNT.

No.	qPCR (Ct)		ELISA	PRNT		Background of Animal						
	ORF3a	N Gene	(OD450)	Neutralization Titer	Species	Sex	Age (yr)	Breed	Province	Source	COVID-19 Patient Owner	Clinical Signs
1	>40	>40	0.6928	1/33	Cat	M	5	Korean short hair	Seoul	Household	No	Fever
2	>40	>40	0.9721	1/24	Cat	F	5	Korean short hair	Seoul	Household	Yes	Fever
3	>40	>40	1.2427	1/45	Cat	F	2	Persian cat	Seoul	Household	No	Cough
4	>40	>40	0.6131	1/33	Cat	M	5	Turkish angora	Seoul	Household	No	Fever
5	>40	>40	0.6762	1/24	Cat	F	3	Korean short hair	Gyeonggi	Household	Yes	Cough/Fever
6	>40	>40	0.9221	1/45	Cat	M	4	Russian blue	Gyeonggi	Household	No	Cough
7	>40	>40	0.6793	1/24	Cat	M	Unknown	Unknown	Gyeonggi	Animal shelter	Yes	Cough/Fever
8	>40	>40	0.6178	>1/5	Cat	F	3	Korean short hair	Gangwon	Household	Yes	Cough/Fever
9	>40	>40	1.2123	1/33	Cat	F	2	Korean short hair	Chungbuk	Household	Yes	Fever
10	>40	>40	1.8163	1/45	Cat	M	7	Korean short hair	Chungbuk	Household	Yes	Fever
11	>40	>40	0.92193	1/33	Cat	F	3	Korean short hair	Chungnam	Household	Yes	Cough
12	>40	>40	0.6532	1/24	Cat	M	1	Korean short hair	Gyeongbuk	Household	No	Cough/Fever
13	>40	>40	1.1262	1/33	Cat	M	Unknown	Korean short hair	Gyeongbuk	Animal shelter	Yes	Fever
14	>40	>40	1.2013	1/33	Cat	F	1	Unkown	Gyeongbuk	Animal shelter	Yes	Cough
15	>40	>40	1.1136	1/33	Cat	M	8	Korean short hair	Gyeongnam	Household	No	Cough
16	>40	>40	1.0252	1/45	Cat	M	7	Persian cat	Jeonbuk	Household	Yes	Cough/Fever
17 *	25.321	25.182	0.5824	1/45	Cat	M	2	Korean short hair	Jeonbuk	Household	Yes	Cough/Fever
18	>40	>40	0.7122	1/33	Cat	F	5	Turkish angora	Jeonnam	Household	Yes	Fever
19	>40	>40	0.9377	1/33	Cat	M	3	Korean short hair	Jeonnam	Household	No	Cough/Fever

* Case where SARS-CoV-2 RNA was detected via qPCR.

**Table 3 viruses-16-01113-t003:** Viral copy number and CPE of JBNU-cat isolate in Vero E6 cells.

Sample	Copy Number (log10)	Cq
Negative	1.285	40
Positive	5.315	21.856
JBNU-cat 1st	5.439	21.3
JBNU-cat 3rd	5.662	20.296

## Data Availability

The data presented in this study are available on request from the corresponding author.

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
