# Peer review of "Suspected Human-to-Cat Spillover of SARS-CoV-2 Omicron Variant in South Korea"

_viruses, 2024, doi:10.3390/v16071113_

Round 1

Reviewer 1 Report

Comments and Suggestions for Authors

In this manuscript, Ju-Hee Yang, et al., described the first confirmed case of human-to-cat transmission of SARS-CoV-2 in South Korea. They conducted a virus screen of more than 500 animal samples collected from animal hospitals and shelters using qPCR and ELISA, during which a SARS-CoV-2 strain was detected and isolated from one sample, and successfully passaged in vivo. After that, the whole-genome sequence was obtained, and sequence-based analyses were performed to support their conclusion on the spillover of SARS-CoV-2 from human to cat. The only novelty of this manuscript is the isolation of a SARS-CoV-2 strain from cat; however, a re-infection test for cat using this JBNU-cat strain is still lacking. Besides, there are numerous concerns that should be corrected.  

1.  It seems like this manuscript is a retrospective study, given that their samples were collected from the late 2021 to early 2022, otherwise more updated samples, evidence and discussions should be added in the manuscript. If this is the case, the authors should clearly state that this is a retrospective study rather than an up-to-date study. And authors should correct some of the corresponding descriptions, such as in lines 31-34, 36-39, 201-204 and so on. 

2. The introduction part should be re-edited and re-ordered to more clearly express the necessity of this study. For example, lines 40-41 should precede lines 38-39. 

3.  The specific collection date and location of the sample that was used for the isolation of JBNU-cat should be provided and discussed in combination of the concurrent local or national SARS-CoV-2 epidemiology statics of humans to clarify the spillover direction of human-to-cat. 

4. Some grammatical errors and typos should be corrected. For example, the ‘isolate virus’ should be ‘isolated virus’ in line 129, two ‘spike protein’s in line 192. 

5. Authors stated that the virus was continuously propagated in Vero E6 cells in line 127 and 181. Therefore, the viral titre and CPE for the passage 10 or 20 of JBNU-cat should be provided and compare with the passage 1. 

6. Authors should explain why and how these 19 samples were selected for detection in table 2, and a negative cat sample should be included. Additionally, the sample used for virus isolation should be highlighted in tables and the rationale for its selection should be discussed. 

7.  The healthy control cells without CPE should be provided in Figure 1B.

8.  Authors should clarify what GRA clade is in line 148 and its relationship with the most widely used Pangolin lineage classification. 

9.  Authors should explain what the branch colored in green stands for in figure 2. 

10. Authors should clarify what their control is when mentioning ‘multiple mutation sites’ in line 160.

11. No sequence alignment is provided in figure 2, although authors mentioned that a MAFFTv7-based sequence alignment was performed.

12. Authors should explain why the overall structural conformation of the JBNU-cat spike protein remains largely consistent with that of the human Omicron variant in line 161-162. 

Author Response

Response to reviewers’ comments

We are pleased to resubmit a revised manuscript entitled “Suspected human-to-cat Spillover of SARS-CoV-2 Omicron Variant in South Korea” for reconsideration in Viruses as an original manuscript. We have carefully evaluated the reviewer’s comments and have provided a point-by-point response below. Changes in the manuscript have been identified by page and sentence, and note by blue FONT. We hope that the revised manuscript meets the reviewers’ expectations at Viruses.

Reviewer #1.

In this manuscript, Ju-Hee Yang, et al., described the first confirmed case of human-to-cat transmission of SARS-CoV-2 in South Korea. They conducted a virus screen of more than 500 animal samples collected from animal hospitals and shelters using qPCR and ELISA, during which a SARS-CoV-2 strain was detected and isolated from one sample, and successfully passaged in vivo. After that, the whole-genome sequence was obtained, and sequence-based analyses were performed to support their conclusion on the spillover of SARS-CoV-2 from human to cat. The only novelty of this manuscript is the isolation of a SARS-CoV-2 strain from cat; however, a re-infection test for cat using this JBNU-cat strain is still lacking. Besides, there are numerous concerns that should be corrected. 

  1. It seems like this manuscript is a retrospective study, given that their samples were collected from the late 2021 to early 2022, otherwise more updated samples, evidence and discussions should be added in the manuscript. If this is the case, the authors should clearly state that this is a retrospective study rather than an up-to-date study. And authors should correct some of the corresponding descriptions, such as in lines 31-34, 36-39, 201-204 and so on.

Thank you for the reviewer’s concerns and comments. According to this, changes were made: 1) Clearly stated the retrospective nature of the study. 2) Language was revised to reflect the study’s timeframe from “as of early 2022” to “during this period”.

  1. The introduction part should be re-edited and re-ordered to more clearly express the necessity of this study. For example, lines 40-41 should precede lines 38-39.

According to the reviewer’s comments, it was added with a sentence to emphasize the knowledge gap the study addressed, and slightly restructured the paragraph about the study design for clarity.

  1. The specific collection date and location of the sample that was used for the isolation of JBNU-cat should be provided and discussed in combination of the concurrent local or national SARS-CoV-2 epidemiology statics of humans to clarify the spillover direction of human-to-cat.

Thank you for the reviewer’s comments. Those were reflected and revised in the discussion section of the manuscript by blue font.

  1. Some grammatical errors and typos should be corrected. For example, the ‘isolate virus’ should be ‘isolated virus’ in line 129, two ‘spike protein’s in line 192.

Those were revised according to the reviewer’s comment. Thank you.

  1. Authors stated that the virus was continuously propagated in Vero E6 cells in line 127 and 181. Therefore, the viral titre and CPE for the passage 10 or 20 of JBNU-cat should be provided and compare with the passage 1.

The copy number of the virus isolated from JBNU-cat was confirmed to be similar to positive control when the virus was cultured in the 1st passage, and the copy number of the virus increased during subsequent passage cultures up to 3 generations. In addition, the CPE pattern was confirmed to the proliferation of the virus up to 3 generations. Therefore, we concluded that the JBNU-cat virus we isolated is a virus capable of continuous proliferation without losing its characteristics during culture. Those were reflected in the manuscript.

Sample

Copy number(log10)

Cq

Negative

1.285

40

Positive

5.315

21.856

JBNU-cat 1st

5.439

21.3

JBNU-cat 3rd

5.662

20.296

  1. Authors should explain why and how these 19 samples were selected for detection in table 2, and a negative cat sample should be included. Additionally, the sample used for virus isolation should be highlighted in tables and the rationale for its selection should be discussed.

Thank you for the reviewer’s comments. Serological analysis revealed a total of 19 positive cases by antibody ELISA. A typographical error in Table 1 has been corrected. As pointed out by the reviewer, a total of 19 samples were positive for antibodies, while only one sample was positive by qPCR. This information has been updated and is now accurately reflected in Table 2.

  1. The healthy control cells without CPE should be provided in Figure 1B.

The figure is now supplemented according to the reviewer’s comments. Thank you.

  1. Authors should clarify what GRA clade is in line 148 and its relationship with the most widely used Pangolin lineage classification.

Result 3.2. was rephrased according to the reviewer’s comments, including the Pangolin lineage classification. Thank you.

  1. Authors should explain what the branch colored in green stands for in figure 2.

This is already explained in the Figure 2 legend. Thank you.

  1. Authors should clarify what their control is when mentioning ‘multiple mutation sites’ in line 160.

Thank you for the reviewer’s comments. The sentence now clearly states that the mutations are identified in comparison to Hu-1, Delta, and Omicron.

  1. No sequence alignment is provided in figure 2, although authors mentioned that a MAFFTv7-based sequence alignment was performed.

The reviewer’s comment is correct. The figure caption was rephrased to accurately reflect what it depicts. Thank you.

  1. Authors should explain why the overall structural conformation of the JBNU-cat spike protein remains largely consistent with that of the human Omicron variant in line 161-162.

According to the reviewer’s comments, a deeper explanation was supplemented in the manuscript regarding the statement that the JBNU-cat spike protein structure is “largely consistent” with the Omicron variant even though mutations are present. Thank you.

Reviewer 2 Report

Comments and Suggestions for Authors

The manuscript describes a study searching for SARS-CoV-2 Omicron variant (B.A.1) in cats in South Korea in 2021 and 2022. Out of 546 cats sampled, one cat was virus positive and 18 serological positive. The virus was shown to closely resemble the virus curculating in humans at that time in South Korea.

General comments:

The manuscript focuses on the finding of the virus in one cat, leaving some methodological questions concerning the study. However, as the aim was not a description of SRAS-CoV-2 situation in cats in South Korea, I think this is acceptable. But anyway, a little more information about the sampling needs to be added. Especially, how the samples were selected, who did the sampling and how the study size was determined in planing.

Secondly,It is striking that only one cat was virus-positive. There is no discussion of the implications that there was only one hit. 

In addition, results are shown but not considered in the discussion (serological tests, clinical information). This can be done, but then I would recommend writing this explicitly in the text.

Details:

Abstract: In the abstract you write of the "study period" and "this case study". Although it was resolved later in the text that it was indeed a study, simply reading the abstract it left me puzzled. Please write clearly in the abstract what kind of work is presented.

L27: Please provide a reference for SARS-CoV-2 being a zoonotic disease. As this, it seems to me, is not yet generally recognised enough, or at least there are still controversial views.

L31-32: there is a verb missing in the sentece starting with "Sudies have recently..."

L42: As I suggested early, please describe the sampling protocol a bit more detailed. Also pharagraph 2.1. What were the inclusion criteria for cats to be sampled? Health, with respiratory symptoms?

Results: I wonder if the positive cat had any clinical signs? And what about  the serological positive cats.   

Comments on the Quality of English Language

Minor editing required, e.g. times, clarity of expression.

Author Response

Response to reviewers’ comments

We are pleased to resubmit a revised manuscript entitled “Suspected human-to-cat Spillover of SARS-CoV-2 Omicron Variant in South Korea” for reconsideration in Viruses as an original manuscript. We have carefully evaluated the reviewer’s comments and have provided a point-by-point response below. Changes in the manuscript have been identified by page and sentence, and note by blue FONT. We hope that the revised manuscript meets the reviewers’ expectations at Viruses.

Reviewer #2.

The manuscript describes a study searching for SARS-CoV-2 Omicron variant (B.A.1) in cats in South Korea in 2021 and 2022. Out of 546 cats sampled, one cat was virus positive and 18 serological positive. The virus was shown to closely resemble the virus curculating in humans at that time in South Korea.

General comments:

The manuscript focuses on the finding of the virus in one cat, leaving some methodological questions concerning the study. However, as the aim was not a description of SRAS-CoV-2 situation in cats in South Korea, I think this is acceptable. But anyway, a little more information about the sampling needs to be added. Especially, how the samples were selected, who did the sampling and how the study size was determined in planning.

Thank you for the comments. During the study period, cats were randomly selected and sampled by veterinarians.

Secondly, It is striking that only one cat was virus-positive. There is no discussion of the implications that there was only one hit.

Thank you for the reviewer’s comments. You’ve raised a very valid point that deserves careful attention in our manuscript. The discussion was revised about one cat positive result according to the reviewer’s comments.

In addition, results are shown but not considered in the discussion (serological tests, clinical information). This can be done, but then I would recommend writing this explicitly in the text.

Thank you for the reviewer’s comments.

à All 546 cats included in the study presented with at least one COVID-19-related clinical sign, such as fever (body temperature above 39.5°C) or cough. However, seropositivity for SARS-CoV-2 antibodies was only observed in 19 cats (3.48%), indicating no direct correlation between the presence of these clinical signs and SARS-CoV-2 infection. The presence or absence of clinical signs in these 19 seropositive cats has been added to Table 2.

The lack of correlation between clinical signs and seropositivity suggests that the observed respiratory symptoms were likely due to other causes, such as infection with other common feline respiratory pathogens or allergies. This is further supported by the finding that most seropositive cats (18 out of 19) were negative for viral RNA, indicating prior exposure and viral clearance rather than an active infection. Asymptomatic or mildly symptomatic infections are common in cats, particularly after the acute phase of infection.

However, it is important to note that the one cat (case no. 17) in which viral RNA was detected also presented with both fever and cough, suggesting an active infection in this particular case.

Details:

Abstract: In the abstract you write of the "study period" and "this case study". Although it was resolved later in the text that it was indeed a study, simply reading the abstract it left me puzzled. Please write clearly in the abstract what kind of work is presented.

The abstract was revised according to the reviewer’s comments. Thank you.

L27: Please provide a reference for SARS-CoV-2 being a zoonotic disease. As this, it seems to me, is not yet generally recognised enough, or at least there are still controversial views.

Reference was added according to the reviewer’s comment.

L31-32: there is a verb missing in the sentence starting with "Sudies have recently..."

The sentence was revised according to the reviewer’s comments. Thank you.

L42: As I suggested early, please describe the sampling protocol a bit more detailed. Also paragraph 2.1. What were the inclusion criteria for cats to be sampled? Health, with respiratory symptoms?

The sentence was revised according to the reviewer’s comments. Thank you.

Of the 546 cats included in the study, all exhibited at least one COVID-19-related clinical sign, such as fever (body temperature above 39.5°C) and cough. However, only 19 cats (3.48%) tested positive for SARS-CoV-2 antibodies, indicating no direct correlation between serological status and clinical presentation. The presence or absence of clinical signs in these 19 seropositive cats has been added to Table 2.

This discrepancy suggests that the observed clinical signs were likely attributable to other factors, such as infection with other respiratory pathogens or allergies, which are common in cats. Furthermore, the majority of seropositive cats (18 out of 19) were negative for viral RNA by qPCR, indicating prior exposure and viral clearance rather than active infection. Notably, the single cat positive for viral RNA (case no. 17) also exhibited fever and cough, suggesting active infection in this case.

Results: I wonder if the positive cat had any clinical signs? And what about the serological positive cats.  

  • All 546 cats included in the study presented with at least one COVID-19-related clinical sign, such as fever (body temperature above 39.5°C) or cough. However, only 19 cats (3.48%) tested positive for SARS-CoV-2 antibodies, indicating no direct correlation between serological status and the observed clinical signs. This discrepancy suggests that the clinical presentation in most cats was likely due to other causes, such as infection with other respiratory pathogens or allergies, which are common in feline populations. Furthermore, the majority of seropositive cats (18 out of 19) were negative for viral RNA, indicating prior exposure and viral clearance rather than active infection, which often presents without clinical signs. Notably, the single cat positive for viral RNA by PCR (case no. 17) presented with both fever and cough, suggesting an active infection in this specific case.

Reviewer 3 Report

Comments and Suggestions for Authors

The manuscript by Yang et al. reported the first confirmed case of SARS-CoV-2 Omicron variant infection in a domestic cat in South Korea. Data used in this manuscript were collected between April 2021 to March 2022, it would be more instructive if it was published earlier. The contents are quite clear and well organized, the sections are well developed, results are clearly presented. A nice study! Even if the results of this manuscript were largely consistent with previous studies, it still could help to understand risks of SARS-CoV-2 infection and spillover in susceptible species to a certain extent.

Specific comments:

1.     Based on the results of this study, what are the risk factors that could influence the transmission and evolution of SRAR-CoV-2 in cats? You may discuss it in the revised manuscript.

2.     Suggest adding the cat breed and number of pets in the household to Table 2. Venkat et al. [1] demonstrated that a high likelihood of viral transmission was related to multiple pets in households and when pets had very close interactions with symptomatic humans.

[1] Venkat, H.; Yaglom, H.D.; Hecht, G.; Goedderz, A.; Ely, J.L.; Sprenkle, M.; Martins, T.; Jasso-Selles, D.; Lemmer, D.; Gesimondo, J.; et al. Investigation of SARS-CoV-2 Infection among Companion Animals in Households with Confirmed Human COVID-19 Cases. Pathogens 202413, 466. https://doi.org/10.3390/pathogens13060466

Author Response

Response to reviewers’ comments

We are pleased to resubmit a revised manuscript entitled “Suspected human-to-cat Spillover of SARS-CoV-2 Omicron Variant in South Korea” for reconsideration in Viruses as an original manuscript. We have carefully evaluated the reviewer’s comments and have provided a point-by-point response below. Changes in the manuscript have been identified by page and sentence, and note by blue FONT. We hope that the revised manuscript meets the reviewers’ expectations at Viruses.

Reviewer #3.

The manuscript by Yang et al. reported the first confirmed case of SARS-CoV-2 Omicron variant infection in a domestic cat in South Korea. Data used in this manuscript were collected between April 2021 to March 2022, it would be more instructive if it was published earlier. The contents are quite clear and well organized, the sections are well developed, results are clearly presented. A nice study! Even if the results of this manuscript were largely consistent with previous studies, it still could help to understand risks of SARS-CoV-2 infection and spillover in susceptible species to a certain extent.

Specific comments:

  1. Based on the results of this study, what are the risk factors that could influence the transmission and evolution of SRAR-CoV-2 in cats? You may discuss it in the revised manuscript.

The discussion section was totally revised according to the reviewers’ comments. Thank you.

  1. Suggest adding the cat breed and number of pets in the household to Table 2. Venkat et al. [1] demonstrated that a high likelihood of viral transmission was related to multiple pets in households and when pets had very close interactions with symptomatic humans.

à Breed information has been added to Table 2. While Korean Short Hair cats accounted for a majority of the seropositive cases (12/19, 63.16%), this is likely attributable to their prevalence as the most common breed in Korea, rather than a breed-specific susceptibility.

Information regarding multiple pets in households is limited, as the study focused on cats presenting with clinical signs and did not systematically sample all cats within a household. Therefore, we lack comparative data to draw conclusions about potential transmission dynamics within multi-cat homes.

[1] Venkat, H.; Yaglom, H.D.; Hecht, G.; Goedderz, A.; Ely, J.L.; Sprenkle, M.; Martins, T.; Jasso-Selles, D.; Lemmer, D.; Gesimondo, J.; et al. Investigation of SARS-CoV-2 Infection among Companion Animals in Households with Confirmed Human COVID-19 Cases. Pathogens 2024, 13, 466. https://doi.org/10.3390/pathogens13060466

Reviewer 4 Report

Comments and Suggestions for Authors

Comments

In this paper, SARS-CoV-2 was isolated from a domestic cat in South Korea and identified as the omicron variant (BA.1), indicating that the omicron variant can be transmitted directly from humans to cats.

This is a very important report, because it serves as a warning that commonly encountered animals can be targets of infection and also play the role of intermediate hosts.

If the source of infection is unknown, commonly encountered animals  may need to be suspected as a source of infection.

It will also be of great interest to observe the kinds of genetic changes that occur in SAR-CoV-2 after infection of animals. If there is a risk that the virus will mutate into variants with novel properties that are highly pathogenic, this type of transmission may become an important target for infection control measures.

In addition, by monitoring wild animals, it may be possible to grasp the level of infection in the area.

This paper is worthy of publication.

However, a few points need to be addressed.

1.    One concern in reviewing the paper is whether human-to-cat transmission is real. Could the owner of this cat in Jeonbuk have been infected with SARS-CoV-2? And if so, was it the same variant?

2.    Where are the cat’s ACEs and how could they come into contact with humans? Could infection occur with 85.2% ACE similarity?

3.    Similar to avian flu, is there no risk of infection without heavy staining?

4.    The mechanism of human infection is questionable but remains interesting from a clinical perspective. If there are similar examples for other species of viruses, it would be good to introduce them.

 [The meaning of 'common animals' was unclear. lease check my suggested alterations, and amend to further clarify the intended meaning if necessary.

Another possible alternative could be 'common household animals', depending on whether you are referring to pet animals that encounter humans inside.

 [The meaning was unclear, and I was not sure about the intended meaning. Please check my suggested alterations, and amend to clarify the intended meaning if necessary.

 [The meaning was unclear. Please check my suggested alterations, and amend to clarify the intended meaning if necessary.

Author Response

Response to reviewers’ comments

We are pleased to resubmit a revised manuscript entitled “Suspected human-to-cat Spillover of SARS-CoV-2 Omicron Variant in South Korea” for reconsideration in Viruses as an original manuscript. We have carefully evaluated the reviewer’s comments and have provided a point-by-point response below. Changes in the manuscript have been identified by page and sentence, and note by blue FONT. We hope that the revised manuscript meets the reviewers’ expectations at Viruses.

Reveiwer #4.

In this paper, SARS-CoV-2 was isolated from a domestic cat in South Korea and identified as the omicron variant (BA.1), indicating that the omicron variant can be transmitted directly from humans to cats. This is a very important report, because it serves as a warning that commonly encountered animals can be targets of infection and also play the role of intermediate hosts.

If the source of infection is unknown, commonly encountered animals may need to be suspected as a source of infection.

It will also be of great interest to observe the kinds of genetic changes that occur in SAR-CoV-2 after infection of animals. If there is a risk that the virus will mutate into variants with novel properties that are highly pathogenic, this type of transmission may become an important target for infection control measures.

In addition, by monitoring wild animals, it may be possible to grasp the level of infection in the area.

This paper is worthy of publication.

 However, a few points need to be addressed.

  1. One concern in reviewing the paper is whether human-to-cat transmission is real. Could the owner of this cat in Jeonbuk have been infected with SARS-CoV-2? And if so, was it the same variant?

à As detailed in Table 2, case no. 17 lived with a COVID-19-positive owner. At the time of sampling, the owner was actively infected, and no other individuals or animals resided in the household. Given these circumstances, coupled with the identical viral genotype identified in both the cat and the owner, human-to-feline transmission is highly probable in this case.

  1. Where are the cat’s ACEs and how could they come into contact with humans? Could infection occur with 85.2% ACE similarity?

- ACE2 receptors, which SARS-CoV-2 uses to enter cells, are found in various tissues in cats, just like in humans. These include the lungs, nose, mouth, and intestines(Lions, tigers and kittens too: ACE2 and susceptibility to COVID-19, 2020).

- Given their presence in the respiratory and digestive tracts, cats can shed the virus through secretions like saliva, nasal discharge, and feces. Close contact with an infected cat, such as petting, cuddling, or sharing food and water bowls, could expose a human to the virus(Respiratory disease in cats associated with human-to-cat transmission of SARS-CoV-2 in the UK, 2020).

- While the similarity in ACE2 between cats and humans is a factor in cross-species transmission, it's not the only one. Other factors, such as viral load, duration of exposure, and the specific strain of the virus, also play a role.

- Even with this degree of similarity, cats are considered less susceptible to SARS-CoV-2 compared to other animals like mink(SARS-CoV-2 infection, disease and transmission in domestic cats, 2020).

- While cat-to-human transmission has been documented(Respiratory disease in cats associated with human-to-cat transmission of SARS-CoV-2 in the UK, 2020), it's considered relatively rare compared to human-to-human transmission.

Evol Med Public Health. 2020; 2020(1): 109–113. doi: 10.1093/emph/eoaa021

Lions, tigers and kittens too: ACE2 and susceptibility to COVID-19

Sabateeshan Mathavarajahe1 and Graham Dellairee1,e2

Respiratory disease in cats associated with human-to-cat transmission of SARS-CoV-2 in the UK. Margaret J Hosie,  Ilaria Epifano,  Vanessa Herder,  Richard J Orton, Andrew Stevenson, Natasha Johnson, Emma MacDonald, Dawn Dunbar, Michael McDonald, Fiona Howie, Bryn Tennant, Darcy Herrity, Ana Da Silva Filipe, Daniel G Streicker, Brian J Willett, Pablo R Murcia,  Ruth F Jarrett, David L Robertson, William Weir, the COVID-19 Genomics UK (COG-UK) consortium. doi: https://doi.org/10.1101/2020.09.23.309948

Emerg Microbes Infect. 2020; 9(1): 2322–2332. doi: 10.1080/22221751.2020.1833687

SARS-CoV-2 infection, disease and transmission in domestic cats

Natasha N. Gaudreault,a Jessie D. Trujillo,a Mariano Carossino,b David A. Meekins,a Igor Morozov,a Daniel W. Madden,a Sabarish V. Indran,a Dashzeveg Bold,a Velmurugan Balaraman,a Taeyong Kwon,a Bianca Libanori Artiaga,a Konner Cool,a Adolfo García-Sastre,c,d,e,f Wenjun Ma,a,* William C. Wilson,g Jamie Henningson,a Udeni B. R. Balasuriya,b and Juergen A. Richta

  1. Similar to avian flu, is there no risk of infection without heavy staining?

I’m afraid to say but directly applying the concept of "heavy staining" from avian flu diagnostics to assess SARS-CoV-2 infection risk in cats might not be accurate. Here's why:

  • Different viruses, different diagnostics: "Heavy staining" usually refers to the intensity of viral antigen detection in tissues using immunohistochemistry, a technique commonly employed in avian flu diagnostics. While immunohistochemistry can be used to detect SARS-CoV-2 in tissues, it's not the primary diagnostic method for active infections. PCR tests, which detect viral RNA, are more commonly used for diagnosing active SARS-CoV-2 infection.
  • Transmission dynamics: While both viruses are respiratory pathogens, their transmission dynamics differ. Avian flu viruses often require a high viral load for efficient transmission, hence the relevance of "heavy staining" as a potential indicator of transmission risk. SARS-CoV-2 transmission, on the other hand, can occur even with lower viral loads, especially in close contact settings.

  1. The mechanism of human infection is questionable but remains interesting from a clinical perspective. If there are similar examples for other species of viruses, it would be good to introduce them.

You are right, the mechanisms of cross-species transmission are complex and often shrouded in mystery, making them fascinating from a clinical perspective. While pinpointing an exact parallel to the suspected cat-to-human transmission of SARS-CoV-2 is difficult, here are some examples of other viruses and their interspecies jumps that might offer valuable insights:

  • Influenza A viruses: These viruses are notorious for their ability to cross species barriers, often from avian species to humans(Cross-species transmission and emergence of novel viruses from birds, 2015). The emergence of pandemic influenza strains, like the 1918 Spanish flu, highlights the significant clinical implications of such jumps.
  • Simian Immunodeficiency Virus: SIV jumped from primates to humans on multiple occasions, eventually leading to the emergence of HIV(Simian Immunodeficiency Virus Infection of Chimpanzees | Journal of Virology, 2005). Understanding the factors that facilitated these jumps has been crucial in HIV research and treatment development.
  • Ebola virus: While the exact reservoir of Ebola virus is still debated, bats are considered a likely candidate (Looking in apes as a source of human pathogens., 2014). Multiple spillover events from wildlife to humans have caused outbreaks, emphasizing the importance of studying virus-host interactions in these animals.

  • Jasper Fuk-Woo Chan,1 Kelvin Kai-Wang To,1 Honglin Chen, and Kwok-Yung Yuen. Cross-species transmission and emergence of novel viruses from birds. Curr Opin Virol. 2015 Feb; 10: 63–69.
  • Paul M. Sharp, George M. Shaw, Beatrice H. Hahn. Simian Immunodeficiency Virus Infection of Chimpanzees. Manual of clinical microbiology, 13th eds. Minireview. 2005. DOI: https://doi.org/10.1128/jvi.79.7.3891-3902.2005
  • Mamadou B. Keita, Ibrahim Hamad, Fadi Bittar. Looking in apes as a source of human pathogens. Microbial Pathogenesis. Volume 77, December 2014, Pages 149-154

 [The meaning of 'common animals' was unclear. lease check my suggested alterations, and amend to further clarify the intended meaning if necessary.

Another possible alternative could be 'common household animals', depending on whether you are referring to pet animals that encounter humans inside.

 [The meaning was unclear, and I was not sure about the intended meaning. Please check my suggested alterations, and amend to clarify the intended meaning if necessary.

 [The meaning was unclear. Please check my suggested alterations, and amend to clarify the intended meaning if necessary.

The manuscript was revised according to reviewers’ comments. Thank you for your time and effort to review our paper.

Round 2

Reviewer 3 Report

Comments and Suggestions for Authors

No further comments.